# The water supply association analysis method in Shenzhen based on kmeans clustering discretization and apriori algorithm

Xin Liu[1,2], Xuefeng Sang[2]*, Jiaxuan Chang[2], Yang Zheng[2], Yuping Han[1]

**1** School of Water Conservancy, North China University of Water Resources and Electric Power, Zhengzhou, China, **2** Research Office for Water Resources Management, China Institute of Water Resources and Hydropower Research, Beijing, China

* sangxf@iwhr.com

**Data Availability Statement:** https://www.protocols.io/workspaces/water-supply-association-analysis.

## Abstract

Since water supply association analysis plays an important role in attribution analysis of water supply fluctuation, how to carry out effective association analysis has become a critical problem. However, the current techniques and methods used for association analysis are not very effective because they are based on continuous data. In general, there is different degrees of monotone relationship between continuous data, which makes the analysis results easily affected by monotone relationship. The multicollinearity between continuous data distorts these analytical methods and may generate incorrect results. Meanwhile, we cannot know the association rules and value interval between features and water supply. Therefore, the lack of an effective analysis method hinders the water supply association analysis. Association rules and value interval of features obtained from association analysis are helpful to grasp cause of water supply fluctuation and know the fluctuation interval of water supply, so as to provide better support for water supply dispatching. But the association rules and value interval between features and water supply are not fully understood. In this study, a data mining method coupling kmeans clustering discretization and apriori algorithm was proposed. The kmeans was used for data discretization to obtain the one-hot encoding that can be recognized by apriori, and the discretization can also avoid the influence of monotone relationship and multicollinearity on analysis results. All the rules eventually need to be validated in order to filter out spurious rules. The results show that the method in this study is an effective association analysis method. The method can not only obtain the valid strong association rules between features and water supply, but also understand whether the association relationship between features and water supply is direct or indirect. Meanwhile, the method can also obtain value interval of features, the association degree between features and confidence probability of rules.

**Funding:** X.S., 2016YFC0401407, National Key Research and Development Program of China and 51679253, National Natural Science Foundation of China; Y.H., 51679089, National Natural Science Foundation of China; X.L., Innovation Foundation of North China University of Water Resources and Electric Power for PhD Graduates.

**Competing interests:** No authors have competing interests

## Introduction

The water supply fluctuation can be affected by a variety of factors, including floating population (FP), rainfall (R), industrial structure, human activities and so on. If we want to understand the cause of the water supply fluctuation, a scientific analysis method need to be designed to carry out the association analysis between features and water supply (WS), so as to grasp fluctuation interval of WS from change in value interval of these features. However, this is not a traditional problem of finding the optimal solution in multiple objectives [1], but a complex problem of association analysis, so the method to find the optimal solution based on the objective function and heuristic algorithm cannot solve this problem [2, 3]. We need to know not only the association rules between features and WS, but how are they correlated to WS, and what are the sensitive interval of features.

For example, an increase in FP will lead to an increase in domestic water use (DWU), industrial water use (IWU) and service industry water use (SIWU), which in turn will lead to an increase in sewage discharge. But the wastewater reuse (WR) can reduce the amount of WS to a certain extent, and we cannot know the fluctuation interval of WS in the end. This situation does not provide a strong support for water supply dispatching. This is because it is found through studies and experiments that the water demand prediction can only accurately predict the trend of water demand, it is difficult to accurately predict the fluctuation. The acute fluctuation of WS causes large errors between the amount of WS and predicted value of water demand. However, the water supply dispatching depends on water demand prediction to a large extent. If the amount of water diversion is greater than the amount of WS, the water level of the reservoirs will rise. Once there is a large-scale rainfall, the reservoir may release surplus water. If the amount of water diversion is less than the amount of WS, the water level of the reservoirs will fall. According to the dispatching regulation of the reservoir, it is possible to increase the amount of water diversion in the next few days in order to raise the water level of the reservoir. If the amount of WS decreases and there is large-scale rainfall, the reservoir will release more surplus water. This will waste a lot of limited water resources and decrease the efficiency of WS. Therefore, it is necessary to grasp the causes and laws of water supply fluctuation so that water supply dispatching can be carried out more scientifically. In the attribution analysis of water supply fluctuation, the association analysis of WS is essential.

Common association analysis methods include similarity analysis [4, 5], cluster analysis [6, 7], regression model [8–10], time series model [11–13], and artificial neural network (ANN) [14–16]. Similarity analysis and clustering analysis methods can effectively classify a variety of data, and these methods have been widely used in many fields [17, 18]. Regression models and time series models can generate multiple regression equations and autoregressive equations, and the coefficients of the equations can represent the association relationship and association degree between independent variables and dependent variable [19–21]. The ANN models can generate potential quantitative relationships between features, which can be transmitted by weights and biases.

However, based on a large number of studies and experiments, for continuous data, it is found that there is serious multicollinearity [22, 23] between features. The regression models are prone to distortion or generate spurious regression equation [24], resulting in invalid association analysis. When an independent variable that has a strong monotone relationship with the dependent variable is added, the previously sensitive independent variable is likely to become less sensitive. Meanwhile, the analysis results of regression models, similarity analysis and clustering analysis are easily affected by monotone relationship [25]. If there is a strong monotone relationship between the features, the features can be easily judged to be similar or clustered into a category. The performance of ANN model depends on the data quality and

input feature selection to a large extent. If the input features and output features of ANN model have a strong monotone relationship, the modeling effect of ANN will be good, and the input features can well simulate the changes of output features. The work in [16] used some multiple feature selection techniques, such as Pearson correlation and principal component analysis (PCA), but the Pearson correlation coefficient is an evaluation method of the linear relationship between two features. When the features do not follow a normal distribution or have more complex linear correlation relationship, the Pearson correlation coefficient is no longer valid. The PCA has mapped the original features, and we can only use these new features to simulate the change of the target feature, but we cannot know the association rules between the original features.

In fact, the multicollinearity and monotone relationship often exist between continuous data, which will have a great impact on the effectiveness of the analysis results. More importantly, there is no means of validating the analysis results, nor can the value interval of features be obtained. If the value interval of features changes, we cannot know whether these features are still sensitive. These facts make these analysis results unreliable. In addition, through the above methods, we still cannot know whether features are directly or indirectly associated, nor can we know their association rules. Therefore, these analysis methods reveal huge limitations in the association analysis, and surprisingly little research has been done on valid association analysis between features and WS.

At present, data mining methods [26–28] have gradually become a major breakthrough in knowledge discovery. The data mining methods can find the hidden information that the data cannot tell us, and obtain the previously unknown and valuable knowledge. Consequently, this study proposed a data mining method coupling discretization [29–31] and apriori algorithm to solve these problems in water supply association analysis. The kmeans clustering algorithm [32, 33] is used to carry out the discretization of continuous data to obtain the one-hot encoding that can be recognized by apriori, and the discretization can also avoid the influence of multicollinearity and monotone relationship. The apriori algorithm [34–36] is used to carry out the association analysis. A scientific method of water supply association analysis was proposed, the strong association rule (SAR) was established, and the value interval between features and confidence probability of the SAR were recognized. The water supply association analysis is helpful to understand the causes of water supply fluctuation and grasp the fluctuation interval of WS, which can provide strong support for urban water supply dispatching.

## Materials and methods

### Discretization

Apriori algorithm is a rule-based machine learning algorithm [37–39], which can effectively find association rules between features. However, the research data are continuous, which is inconsistent with the data structure of apriori algorithm. The apriori algorithm can only identify one-hot encoding, which is one-bit valid encoding. Therefore, it is necessary to carry out data discretization and transform the encoding into one-hot encoding, which is the data structure that can be recognized by apriori. The data discretization is to divide the value range of data into discrete intervals, and discrete data can also avoid the influence of multicollinearity and monotone relationship.

The quality of data discretization will have an impact on the association rules and value interval generated by the apriori algorithm. The data discretization methods include supervised methods and unsupervised methods, and the classification criterion is whether the data contains category information. The supervised discretization methods need to takes into account category information, while unsupervised discretization methods do not. If the

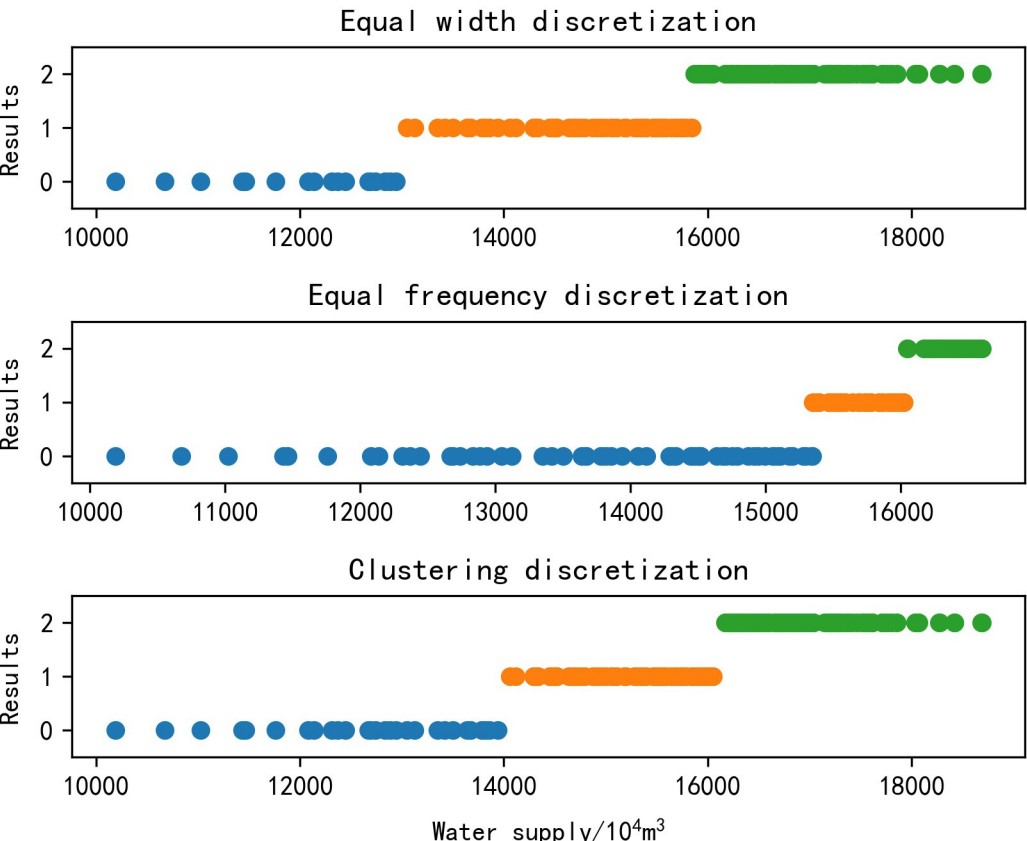

**Fig 1. Comparison of discretization methods.** 0, 1 and 2 are the category label.

supervised discretization method is to be used, the category of the target data needs to be manually annotated. Therefore, unsupervised discretization methods are used more widely, and the unsupervised discretization methods are used in this study. The unsupervised discretization methods include equal width discretization, equal frequency discretization and clustering discretization. This study shows the discretization results of three methods (Fig 1), and the kmeans clustering algorithm is used in this study. This is because among the current clustering algorithms, the kmeans clustering algorithm is considered to be a method with good performance and low calculation load, and the kmeans clustering algorithm has strong self-adaptive ability.

As can be seen intuitively from Fig 1, the data is divided into three intervals after discretization, and the blue, orange and green dots represent the value distribution within the three intervals. The value range of WS is [10189, 18687] $10^4 m^3$. Equal width discretization is to divide the value range of data into intervals of the same width. The uneven data distribution results in very few data in the first interval, which can cause damage in the process of data mining. Equal frequency discretization means that the amount of data in each interval is the same. In order to ensure the consistency of quantity, if multiple same data appears, the probability of these data being divided into different intervals is very high, which will also cause damage to the data mining process to a certain extent. At the same time, the equal frequency discretization can also cause the difference of the interval width. However, the interval width of kmeans clustering discretization has little difference, and the data distribution is even, so kmeans

algorithm has a better performance on the data discretization. The kmeans clustering algorithm is finally applied for data discretization in this study.

## Apriori algorithm

Apriori adopts layer by layer iterative search, and the whole process consists of splicing and pruning. The purpose of splicing is to complete the construction of data item set. Then, support degree of data item is calculated. If an item does not meet the support degree threshold, it is pruned off, and we can obtain frequent item set through splicing. The association rules are generated through the cross of frequent item and those rules that do not meet the confidence degree threshold are pruned off. The valid association rules are obtained after the lift degree validation. The apriori algorithm iteratively generates association rules and validates them until it stops when no more rules can be generated. The apriori algorithm uses recursive method, and the detailed algorithm process (Eqs 1 to 4) are as follows.

Step 1: Initializing k = 1.The candidate set ($CF_k$) of k-item frequent set is calculated, and the frequent set ($SF_k$) that is larger than the support degree threshold is selected.

Step2: $SF_k$ itself crosses to generate the candidate set ($CR_k$) of k-item association rule, and the association rule ($R_k$) that is larger than the confidence degree threshold is selected.

Step3: If the (k+1)-item frequent set is empty, the k-item frequent set is set as the final result and the iteration stops.

$$CF_k = \{i_1, i_2, \cdots, i_n\} \tag{1}$$

$$SF_k = \{c \in CF_k | c.sup \geq sup_{min}\} \tag{2}$$

$$CR_k = Apriori_{gen}(SF_k) \tag{3}$$

$$R_k = \{c \in CR_k | c.conf \geq conf_{min}\} \tag{4}$$

where $i$ is the data item, $c$ is candidate, $sup$ is support degree, $conf$ is confidence degree, $sup_{min}$ is the support degree threshold, $conf_{min}$ is the confidence degree threshold.

Support degree (Eqs 5 and 6) is equivalent to the probability, which represents the ratio between the number of event sample and the number of total sample. Support degree reveals the probability that event occurs. If the $P(A)$ is small, it means that the sample number of event $A$ is small. If $P(AB)$ is small, it can only indicate that the sample number of event $A$ and $B$ is small, but it does not mean that the association degree between event $A$ and $B$ is not strong. The event $B$ may appear in all records where event $A$ appears, which indicates that event $A$ and $B$ are associated, simply because the $N_{total}$ is very large, resulting in a small value of $P(AB)$.

$$sup_A = P(A) = \frac{N_A}{N_{total}} \tag{5}$$

$$sup_{AB} = P(AB) = \frac{N_{AB}}{N_{total}} \tag{6}$$

where $A$ and $B$ are the event, $P$ is the probability, $N$ is the number of sample.

Confidence degree (Eq 7) is equivalent to the conditional probability. If the confidence degree is high, the occurrence probability of $B$ is high when $A$ occurs. On the contrary, the

occurrence probability of *B* is low when *A* occurs.

$$conf_{A \to B} = P(B|A) = \frac{Sup_{AB}}{Sup_A} \tag{7}$$

where *conf* is confidence degree, *A*→*B* is association rule, and *P*(*B*|*A*) is the conditional probability.

The lift degree (Eq 8) is used to validate the SAR and prune off the spurious SAR. From the properties of conditional probability, lift degree is greater than or equal to 1. When lift degree equals to 1, *P*(*AB*) = *P*(*A*)*P*(*B*), then *A* and *B* are independent of each other. This phenomenon indicates that even if the confidence degree is very high, but the lift degree is less than or equal to 1, indicating that this rule is spurious SAR. When the lift degree is greater than 1, the SAR is a valid SAR. The higher the lift degree of SAR, the stronger the association degree of SAR, and the smaller the lift degree of SAR, the weaker the association degree of SAR.

$$lift_{A \to B} = \frac{Conf_{A \to B}}{Sup_B} = \frac{P(B|A)}{P(B)} = \frac{P(AB)}{P(A)P(B)} \tag{8}$$

where *lift* is lift degree.

Another validation standard of apriori algorithm is leverage rate (Eq 9), which is a variant of the lift degree. When the lift degree is less than 1, the leverage ratio is less than 0. When the lift degree is 1, the leverage ratio is 0. When the lift degree is greater than 1, the leverage ratio is greater than 0. Only the calculation methods of the two standards are different, so the leverage ratio is not calculated in this study.

$$leve_{A \to B} = P(AB) - P(A)P(B) \tag{9}$$

where *leve* is leverage rate.

## Coupling method

The different discretization parameters will generate different value intervals. Therefore, this study inputs different discretization parameters into kmeans, hoping to find the optimal discretization parameter (D) through finite iteration to achieve the best effect of data mining.

The purpose of this study is not only to obtain more SAR, but also to obtain valid SAR and detailed value interval. We may lose quality if we focus only on quantity of SAR, and we may lose quantity if we focus only on quality of SAR. Therefore, the objective function (Eq 10) set in this study is that the average lift degree of the valid SARs is maximum. The objective function can avoid the contingency results and make the analytical results of iterative calculation more reliable.

$$objective\ function = \max_D \frac{1}{n} \sum_{i=1}^{n} lift_i \tag{10}$$

where *n* is the number of valid SAR.

If the support degree threshold is set too high, although it can reduce the time to calculate frequent item set in the process of data mining, it is easy to cause some important frequent items hidden in the data to be filtered out. Because the confidence degree need to be calculated after support degree, so the support degree threshold should be set as small as possible. If the confidence degree threshold is set too low, a large number of rules with weak association degree with WS may be generated, leading to a high calculation load and greatly increasing the time of data mining. Since the lift degree of SAR need to be calculated, so the confidence degree threshold does not have to be set very high so as to ensure that more rules are

generated. Therefore, considering the calculation load and efficiency of algorithm, combined with previous data mining experience, the support degree threshold is set to 0.08, and the confidence degree threshold is set to 0.5 (Eq 11). The SAR is the association rule extracted by the support degree threshold and confidence degree threshold, and the SAR validated by the lift degree threshold is the valid SAR.

$$sup_{min} \geq 0.08, conf_{min} \geq 0.5, lift_{min} > 1 \qquad (11)$$

where $lift_{min}$ is the lift degree threshold.

Fig 2 shows the flow chart of coupling method. The D is initialized, and the discretization breakpoints and value intervals are generated to build a data structure that can be recognized by the apriori algorithm. Finally, the SAR can be obtained, and the cross rules in the same category and different categories can be generated and their confidence probabilities and value intervals are obtained. In addition, the method can recognize whether they are directly or indirectly associated. The methods and algorithms in this study are developed using Python 3.

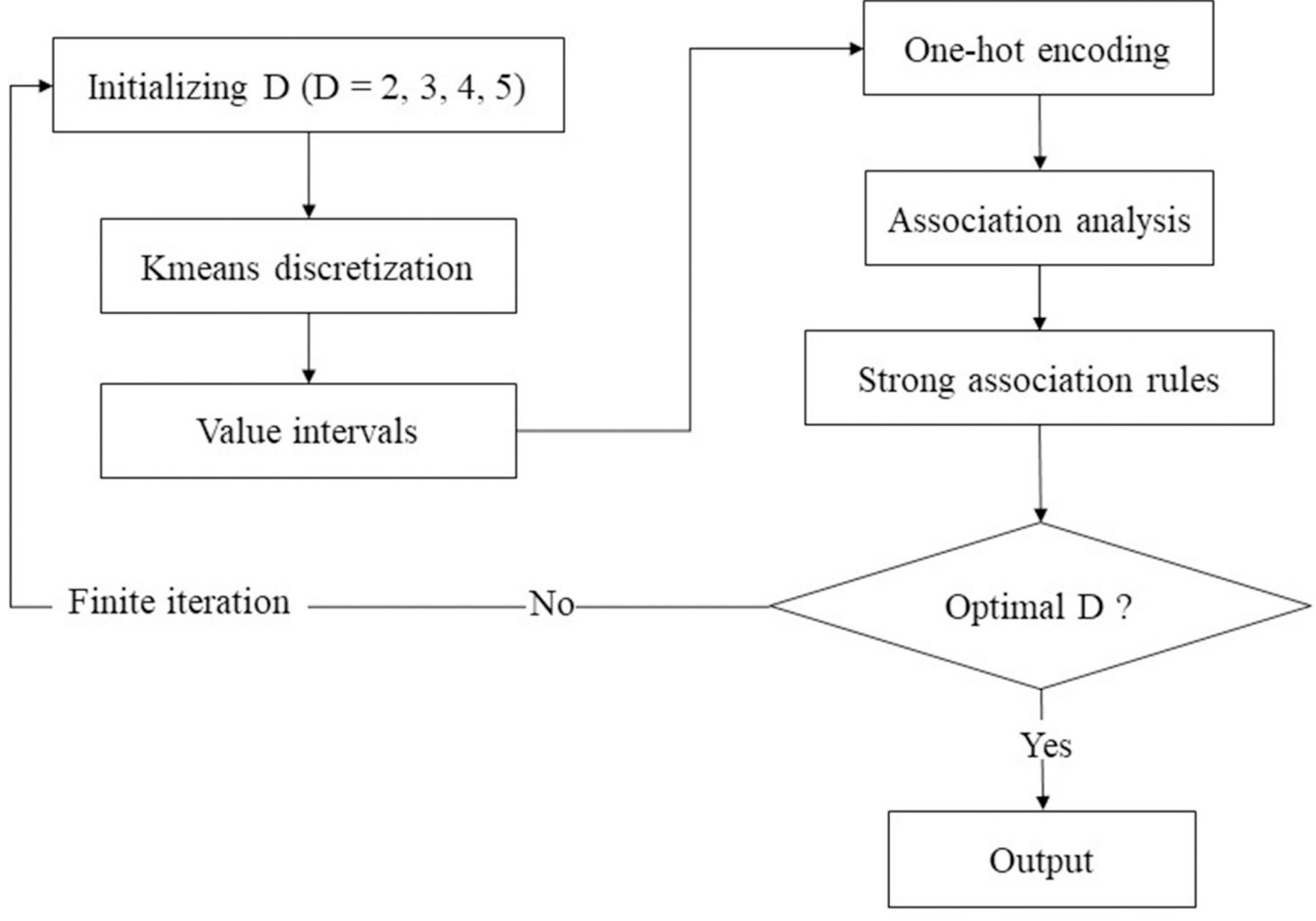

**Fig 2. The flow chart of coupling method.**

## Study area and data description

### Study area

Shenzhen, a sub-provincial city of Guangdong Province, is a special economic zone of China approved by the State Council, adjacent to Hong Kong. Affected by topography, the city has no large rivers, and the water supply reservoirs in Shenzhen are mainly medium reservoirs and small-scale reservoirs. Shenzhen has abundant rainfall, with an average annual rainfall of 1830 mm. But the water supply dispatching in Shenzhen mainly relies on water diversion, which accounts for more than 80 percent of the total water supply. At the same time, there is an acute fluctuation in water supply of Shenzhen. Therefore, it is essential to carry out association analysis of water supply in Shenzhen and understand the fluctuation cause and fluctuation interval of water supply.

### Data description

The water use of Shenzhen consists of DWU, IWU, SIWU, ecological water use, agricultural water use and construction industry water use. The Pearson correlation coefficient is an evaluation method of the linear relationship between two variables. When the variables do not follow a normal distribution or have more complex linear correlation relationship, the Pearson correlation coefficient is no longer valid. Spearman's rank correlation coefficient (SRCC) (Eqs 12 and 13) [40] evaluates a monotone relationship between two variables. SRCC does not require prior knowledge, and its application scope is wider than Pearson correlation coefficient. Therefore, SRCC was applied in this study to calculate the correlation coefficient, and the monotone relationship between DWU, IWU, SIWU and WS was better.

$$d = \sum_{i=1}^{n} |R(X_i) - R(Y_i)|^2 \tag{12}$$

$$SRCC = 1 - \frac{6d}{n(n^2 - 1)} \tag{13}$$

where $R(X)$ and $R(Y)$ are the rank of features $X$ and $Y$; $n$ is the number of element.

In addition, Shenzhen is a fully urbanized city with a FP of more than 8 million. At the same time, a large amount of water use has produced a large amount of wastewater, so the amount of wastewater reuse (WR) in Shenzhen is increasing. The migration of FP, the amount of R and the amount of WR have great influence on WS. Therefore, the six features including DWU, IWU, SIWU, FP, WR and R were selected in this study.

The data for this study came from the monthly data of Shenzhen Bureau of Water, the Shenzhen Bureau of Statistics, the Shenzhen Water Group and Digital Water System from 2004 to 2019. The variable descriptions are presented in Table 1. According to the standard deviation, the fluctuation of WS is acute, and the amount of WR has increased obviously.

**Table 1. Variable description.**

| Feature | Minimum | Maximum | Average value | Standard deviation |
|---|---|---|---|---|
| DWU/$10^4$m$^3$ | 4283.1 | 7107.69 | 5790.65 | 568.46 |
| IWU/$10^4$m$^3$ | 2689.23 | 6066.46 | 4500.22 | 681.92 |
| SIWU/$10^4$m$^3$ | 1663.95 | 5549.5 | 3491.74 | 914.66 |
| FP/$10^4$people | 348.88 | 856.12 | 618.38 | 108.12 |
| R/mm | 30.07 | 398.86 | 158.43 | 110.38 |
| WR/$10^4$m$^3$ | 0.33 | 1696.78 | 574.17 | 566.5 |
| WS/$10^4$m$^3$ | 10189.05 | 18686.78 | 15721.55 | 1682.78 |

## Results

### Exploratory data analysis

The exploratory data analysis (EDA) [41] of the dataset is carried out first. The EDA can let us intuitively understand the characteristics of data and know the potential quantitative relationship between them, which is a valid method of data analysis. Meanwhile, EDA helps discover what the data is trying to tell us and can be used to look for patterns and relationships. Among the various EDA methods, the most effective method is the scatter density plot matrix (Fig 3). The scatter density plot matrix can describe a variety of complex relationships, such as the density distribution of the single feature, the density relationship and correlation relationship between features, and the function of this method is strong. In order to draw the plot clearly, data normalization is performed (Eq 14). However, in the process of data discretization and data mining, the data normalization is not performed to preserve the original potential relationships between the features.

$$MinMaxScaler = \frac{V_i - V_{min}}{V_{max} - V_{min}} \qquad (14)$$

**Fig 3. Scatter density plot matrix.**

where $V_{min}$ and $V_{max}$ are the minimum and maximum values of the sample, and $V_i$ is value at time $i$.

Fig 3 shows that the upper triangle is the scatter distribution plot between two features, the diagonal is the density curve of the single feature, and the lower triangle is the two-dimensional density plot between two features. The scatter distribution plot is used to describe the correlation relationship between two features. According to the scatter distribution plot between FP and WS, when FP increases, WS increases, which indicates that there is a positive correlation relationship between FP and WS. Similarly, the FP is positively correlated with DWU and SIWU, and the WS is positively correlated with the DWU and SIWU. The WS also is positively correlated with the IWU in the interval of [0, 0.5]. But we have no way of knowing how WR and R correlate to other features.

The density curve presents the density distribution of the single feature. The horizontal axis represents the value range of this feature, and the vertical axis represents the density. The WR and R have two wave peaks, all other features have only one wave peak. The wave peak represents the position with the highest density in the value range. The FP is taken as an example. The FP has only one wave peak, indicating that in the value range of FP, only one value interval has a high density. The position with the highest density corresponds to coordinate of approximately 0.5 on the horizontal axis. The value range of FP is [348.88, 856.12] $10^4$people, and the original value of 0.5 on the horizontal axis is 602.5 $10^4$people. The two ends of the value range are the least dense. Similarly, the WR and R show two wave peaks, indicating that these two features have two high-density intervals.

The two-dimensional density plot is used to describe the density between two features. As can be seen from the two-dimensional density plot, the density distribution of features is not even, but denser in some intervals. The two-dimensional density plot between WS and FP is taken as an example. The two-dimensional density plot is concentrated around the line $y = x$, which indicates that there is a good positive correlation relationship between the two features. Meanwhile, when FP is within the interval [0.3,0.7] and WS is within the interval [0.5,1], the color of the density plot is the darkest. This result indicates that when FP is within the interval [0.3,0.7], WS has the maximum density within the interval [0.5,1]. Similarly, according to the two-dimensional density plot between WR and SIWU. Since WR has two high-density intervals, the two-dimensional density plot is divided into two regions. The lower part of the density plot is darker in color, indicating that when WR is within the interval [0,0.5], SIWU has the maximum density within the interval [0.3,0.7].

These results reveal a potential quantitative relationship between features. However, these relationships are still a little fuzzy, and the value interval of feature is not clear enough. So more detailed rules and value intervals still need to be generated by data mining algorithm.

## Experiment results

The calculation results of the coupling method show that the average lift degree of the valid SARs is maximum when D is equal to 3. The larger the value of D, the less the number of valid SAR, and the lower the sensitivity between the features and WS. The smaller the value of D, the more the number of valid SAR, and the higher the sensitivity between the features and WS. At the same time, if the value of D is relatively small, the interval width will be relatively large, and spurious SARs that do not meet the lift degree can appear in the results. The results of D = 3 and 4 are presented in Table 2. When D is equal to 3, the number of valid SAR is 23, the objective function value is 2.71, and the feature is more sensitive, but spurious SARs appear in the results. When D is equal to 4, the number of valid SAR is 20, the objective function value is 2.17, and the sensitivity of features decreases, but there are no spurious SARs in the results.

**Table 2. The result comparison of the different D.**

| D | Objective function value | Number of valid SAR | Whether there is spurious SAR | Feature sensitivity |
|---|---|---|---|---|
| 3 | 2.71 | 23 | Yes | Strong |
| 4 | 2.17 | 20 | No | Weak |

In order to compare the data mining results of different D more clearly, the results of D = 3 and 4 are compared and analyzed in this study. Their discretization results are presented in Tables 3 and 4. The discretization breakpoints are generated within the valid value range of continuous data, and the breakpoints of different D are also different. Compared with the results of D = 3, the results of D = 4 have four discretization intervals and smaller interval width, the interval is divided in more detail.

It was found through experiments that when the number of frequent item is greater than two items, there is almost certainly a strong association degree between features and WS. This is because the feature strongly associated with WS has dominated the SAR, which is equivalent to give this feature a large weight. If a new feature which has a strong association degree with WS is added to the two-item SAR, the three-item SAR will undoubtedly be has a strong association degree with WS. Even if a feature that is completely uncorrelated with WS is added to the two-item SAR, the three-item SAR still has a strong association degree with WS due to the existence of the feature which has a large weight. Therefore, this study only focuses on the one-item SAR and two-item SAR, which can also reduce the calculation load of data mining.

The data mining results of D = 3 are shown in Tables 5 and 6. The SAR are sorted in order of lift degree from highest to lowest. The letter inside the braces represents the feature, and the number outside the braces represents the category to which the feature belongs. This category represents a detailed value range of the feature. The first rule in Table 5 is taken as an example, the FP and WS are directly associated. When the FP is in category 1 (FP is within the interval (348.88, 509.95] $10^4$people) and the WS is in category 1 (WS is within the interval (10189.05, 13878.35] $10^4\text{m}^3$), the support degree, confidence degree and lift degree of SAR are 0.13, 0.71 and 4.73 respectively, showing the strongest association degree. As can be seen from rule 11–14, the lift degree of these SARs is less than 1, revealing that they are spurious SAR and should be discarded.

The FP, DWU, SIWU and WS in the SARs belong to the same category (Table 5). The FP has the strongest association degree with WS, followed by DWU and SIWU, and they are all in category 1. This indicates that when the category of features belongs to the first category, the influence of FP on WS is higher than that of DWU and SIWU. According to rules 7–9, the three SARs belong to the second category, but association degree between SIWU and WS exceeds that between DWU and WS. In the second category, FP still has the strongest

**Table 3. The discretization results of D = 3.**

| Feature | Category and interval of D = 3 | | |
|---|---|---|---|
| | 1 | 2 | 3 |
| DWU/$10^4\text{m}^3$ | (4283.1, 5261.45] | (5261.45, 6074.3] | (6074.3, 7107.69] |
| IWU/$10^4\text{m}^3$ | (2689.23, 4154.58] | (4154.58, 4685.48] | (4685.48, 6066.46] |
| SIWU/$10^4\text{m}^3$ | (1663.95, 2621.15] | (2621.15, 3958.37] | (3958.37, 5549.5] |
| FP/$10^4$people | (348.88, 509.95] | (509.95, 676.51] | (676.51, 856.12] |
| R/mm | (30.07, 118.32] | (118.32, 179.92] | (179.92, 398.86] |
| WR/$10^4\text{m}^3$ | (0.33, 491.92] | (491.92, 618.26] | (618.26, 1696.78] |
| WS/$10^4\text{m}^3$ | (10189.05, 13878.35] | (13878.35, 16709.5] | (16709.5, 18686.78] |

**Table 4. The discretization results of D = 4.**

| Feature | Category and interval of D = 4 | | | |
|---|---|---|---|---|
| | 1 | 2 | 3 | 4 |
| DWU/$10^4$m$^3$ | (4283.1, 5026.03] | (5026.03, 5750.3] | (5750.3, 6311.27] | (6311.27, 7107.69] |
| IWU/$10^4$m$^3$ | (2689.23, 3685.71] | (3685.71, 4538.36] | (4538.36, 4768.16] | (4768.16, 6066.46] |
| SIWU/$10^4$m$^3$ | (1663.95, 2468.94] | (2468.94, 3195.45] | (3195.45, 4590.63] | (4590.63, 5549.5] |
| FP/$10^4$people | (348.88, 475.54] | (475.54, 601.22] | (601.22, 731.63] | (731.63, 856.12] |
| R/mm | (30.07, 94.29] | (94.29, 151.94] | (151.94, 207.23] | (207.23, 398.86] |
| WR/$10^4$m$^3$ | (0.33, 494.99] | (494.99, 524.72] | (524.72, 632.22] | (632.22, 1696.78] |
| WS/$10^4$m$^3$ | (10189.05, 12928.82] | (12928.82, 15712.46] | (15712.46, 17393.34] | (17393.34, 18686.78] |

association degree with WS, revealing that the influence of FP on WS is higher than that of SIWU and DWU. According to rule 4–6, the three SARs belong to the third category, but association degree between SIWU and WS exceeds that between FP and WS. The influence of features on WS from high to low is SIWU, FP and DWU. These results reveals that the association degree between features and WS is affected by the category to which the feature belongs. It can be seen that when the category of features become larger, the SIWU becomes more sensitive to the WS. The higher the category SIWU belongs to, the stronger the association degree between SIWU and WS. For the FP, the SAR between FP and WS has the strongest association degree when the FP belongs to the first category and the second category. Specifically, the WS is most easily affected by FP when WS is within the interval (10189.05, 16709.5] $10^4$m$^3$, and the WS is most easily affected by SIWU when WS is within the interval (16709.5, 18686.78] $10^4$m$^3$. Although the association degree between SIWU and WS in the third category exceeds association degree between FP and WS, the gap of association degree is not large. This result indicates that FP has a large influence on WS in the whole value range of WS. Similarly, the DWU has the largest influence on WS when DWU is within the interval (4283.1, 5261.45] $10^4$m$^3$. Therefore, the more FP, SIWU and DWU, the more WS. There is a positive association relationship between FP, SIWU, DWU and WS. The results show that the relationship between features found by the scatter density plot matrix is consistent with the results of

**Table 5. The one-item SAR of D = 3.**

| Number | SAR | Support degree | Confidence degree | Lift degree |
|---|---|---|---|---|
| 1 | {FP}1→{WS}1 | 0.13 | 0.71 | 4.73 |
| 2 | {DWU}1→{WS}1 | 0.12 | 0.66 | 4.35 |
| 3 | {SIWU}1→{WS}1 | 0.09 | 0.58 | 3.84 |
| 4 | {SIWU}3→{WS}3 | 0.26 | 0.86 | 2.81 |
| 5 | {FP}3→{WS}3 | 0.22 | 0.81 | 2.64 |
| 6 | {DWU}3→{WS}3 | 0.21 | 0.68 | 2.21 |
| 7 | {FP}2→{WS}2 | 0.44 | 0.81 | 1.49 |
| 8 | {SIWU}2→{WS}2 | 0.43 | 0.81 | 1.49 |
| 9 | {DWU}2→{WS}2 | 0.38 | 0.74 | 1.38 |
| 10 | {WR}3→{WS}2 | 0.27 | 0.62 | 1.04 |
| 11 | {R}3→{DWU}2 | 0.28 | 0.53 | 0.94 |
| 12 | {R}3→{FP}2 | 0.29 | 0.56 | 0.92 |
| 13 | {R}3→{WS}2 | 0.28 | 0.53 | 0.88 |
| 14 | {R}3→{SIWU}2 | 0.26 | 0.50 | 0.83 |

**Table 6. The two-item SAR of D = 3.**

| Number | SAR | Support degree | Confidence degree | Lift degree |
|---|---|---|---|---|
| 1 | {FP}1 and {DWU}1→{WS}1 | 0.11 | 0.91 | 6.04 |
| 2 | {FP}1 and {IWU}1→{WS}1 | 0.10 | 0.90 | 5.99 |
| 3 | {FP}1 and {SIWU}1→{WS}1 | 0.09 | 0.82 | 5.42 |
| 4 | {FP}3 and {SIWU}3→{WS}3 | 0.22 | 0.98 | 3.18 |
| 5 | {FP}3 and {DWU}3→{WS}3 | 0.18 | 0.95 | 3.08 |
| 6 | {R}1 and {SIWU}3→{WS}3 | 0.08 | 0.82 | 2.34 |
| 7 | {FP}2 and {IWU}2→{WS}2 | 0.12 | 0.96 | 1.77 |
| 8 | {FP}2 and {SIWU}2→{WS}2 | 0.35 | 0.85 | 1.57 |
| 9 | {FP}2 and {DWU}2→{WS}2 | 0.30 | 0.81 | 1.50 |
| 10 | {R}3 and {SIWU}2→{WS}2 | 0.23 | 0.87 | 1.43 |
| 11 | {WR}3 and {DWU}2→{WS}2 | 0.19 | 0.85 | 1.43 |
| 12 | {WR}3 and {SIWU}2→{WS}2 | 0.21 | 0.82 | 1.39 |
| 13 | {R}3 and {DWU}2→{WS}2 | 0.20 | 0.71 | 1.17 |

data mining. The association degree of features is not completely unchanged, and there is different association degree in different categories.

Although the SAR between R and DWU, FP, WS, SIWU have passed validation of the support degree threshold and confidence degree threshold, the lift degree of these SARs is all less than 1, indicating that they are all spurious SAR (Table 5). The results mean R does not have valid one-item SAR. However, the SAR between WR and WS meet the lift degree threshold, so this SAR is a valid SAR. But the association degree of this SAR is the lowest and they do not belong to the same category. It can be seen that the more WR, the less WS, so there is a negative association relationship between them.

According to Table 6, eight SARs include FP, which reveals that FP also has a strong association degree with WS in two-item SARs. The lift degree of the first SAR in Table 5 is 4.73. As can be seen from rules 1–3 in Table 6, all the features belong to the first category, and the lift degree of the three SARs is greater than 4.73. This phenomenon reveals that the two-item SARs including FP have higher lift degree than the one-item SAR including only FP. It can be seen from rule 1–3 in Table 6, in the first category, FP is more likely to affect WS through affecting DWU, IWU and SIWU, among which FP is most likely to affect WS through affecting DWU. The similar results are found in the second category and the third category. In the second category, the lift degree of SAR between FP and WS in Table 5 is 1.49, and the lift degree of the rules 7–9 in Table 6 is greater than 1.49. Although FP still affects WS through affecting DWU, IWU, and SIWU, FP is most likely to affect WS through affecting IWU. In the third category, the lift degree of SAR between SIWU and WS in Table 5 is 2.81, and the lift degree of the rules 4 and 5 in Table 6 is greater than 2.81, so the FP is most likely to affect WS by affecting SIWU. However, in the third category, the SAR between FP, IWU and WS does not appear in Table 6, which indicates that when WS belongs to the third category, the impact of FP on WS has little association with IWU. Meanwhile, based on the confidence degree of SAR, the value interval of WS can be inferred. For example, according to the rule 4 in Table 6, when the FP is within (676.51, 856.12] $10^4$people and the SIWU is within (3958.37, 5549.5] $10^4$m$^3$, it can be inferred that the probability that WS is within (16709.5, 18686.78] $10^4$m$^3$ is 0.98.

Although there is no valid SAR between R and WS in the one-item SARs, it can be seen from rule 6, 10 and 13 in Table 6 that the R is indirectly associated with WS through SIWU and DWU. But they do not belong to the same category, so there is a negative association

**Table 7. The one-item SAR of D = 4.**

| Number | SAR | Support degree | Confidence degree | Lift degree |
|---|---|---|---|---|
| 1 | {SIWU}4→{WS}4 | 0.09 | 0.69 | 5.11 |
| 2 | {FP}4→{WS}4 | 0.09 | 0.53 | 3.91 |
| 3 | {FP}2→{WS}2 | 0.29 | 0.79 | 2.29 |
| 4 | {SIWU}2→{WS}2 | 0.20 | 0.72 | 2.10 |
| 5 | {DWU}2→{WS}2 | 0.23 | 0.68 | 1.97 |
| 6 | {FP}3→{WS}3 | 0.30 | 0.78 | 1.78 |
| 7 | {SIWU}3→{WS}3 | 0.33 | 0.74 | 1.69 |
| 8 | {DWU}3→{WS}3 | 0.22 | 0.63 | 1.45 |
| 9 | {WR}4→{WS}3 | 0.21 | 0.53 | 1.20 |

relationship between them. This shows that the less R, the more WS, and the more R, the less WS. When the R belongs to the first category, it affects WS through affecting SIWU. When R belongs to the third category, it affects WS through affecting SIWU and DWU, and it is more likely to affect WS by affecting SIWU. Similarly, the WR affects WS through affecting DWU and SIWU, and it is more likely to affect WS by affecting DWU.

Tables 7 and 8 show the data mining results of D = 4. It's similar to the results of D = 3, the FP, SIWU, DWU and WS in SARs belong to the same category, while the R, WR and WS belong to the different categories. There is a positive association relationship between FP, SIWU, DWU and WS, and there is a negative association relationship between WR, R and WS. However, although there is not much difference between the maximum lift degree in Table 5 and the maximum lift degree in Table 7, there is a big gap between the maximum lift degree in Table 6 and the maximum lift degree in Table 8.

As can be seen from Table 7, in the second and third categories, the order of SAR of the features remains unchanged, indicating that the sensitivity of the features to WS decreases. In addition, when the feature belongs to the fourth category, there is no valid SAR between DWU and WS. This result indicates that the DWU has little influence on WS when WS is within (17393.34, 18686.78] $10^4 m^3$, the results of rule 1–2 indicate that WS is mainly affected by SIWU and FP, and the association degree between SIWU and WS is the strongest. There is a valid SAR between WR and WS, and the association degree of this SAR is weakest. But there is no spurious SAR.

According to the Table 8, its results are similar to the results in Table 6. In the second category, the FP affects WS through affecting IWU, DUW and SIWU, and FP is most likely to

**Table 8. The two-item SAR of D = 4.**

| Number | SAR | Support degree | Confidence degree | Lift degree |
|---|---|---|---|---|
| 1 | {FP}2 and {IWU}2→{WS}2 | 0.11 | 0.95 | 2.78 |
| 2 | {FP}2 and {DWU}2→{WS}2 | 0.19 | 0.93 | 2.69 |
| 3 | {FP}2 and {SIWU}2→{WS}2 | 0.18 | 0.78 | 2.26 |
| 4 | {R}1 and {SIWU}3→{WS}3 | 0.17 | 0.86 | 1.98 |
| 5 | {FP}3 and {SIWU}3→{WS}3 | 0.24 | 0.84 | 1.91 |
| 6 | {FP}3 and {DWU}3→{WS}3 | 0.17 | 0.80 | 1.83 |
| 7 | {R}1 and {DWU}3→{WS}3 | 0.10 | 0.77 | 1.76 |
| 8 | {R}4 and {SIWU}3→{WS}3 | 0.14 | 0.68 | 1.54 |
| 9 | {R}4 and {DWU}3→{WS}3 | 0.11 | 0.56 | 1.29 |
| 10 | {WR}4 and {SIWU}3→{WS}3 | 0.17 | 0.87 | 1.98 |
| 11 | {WR}4 and {DWU}3→{WS}3 | 0.10 | 0.83 | 1.90 |

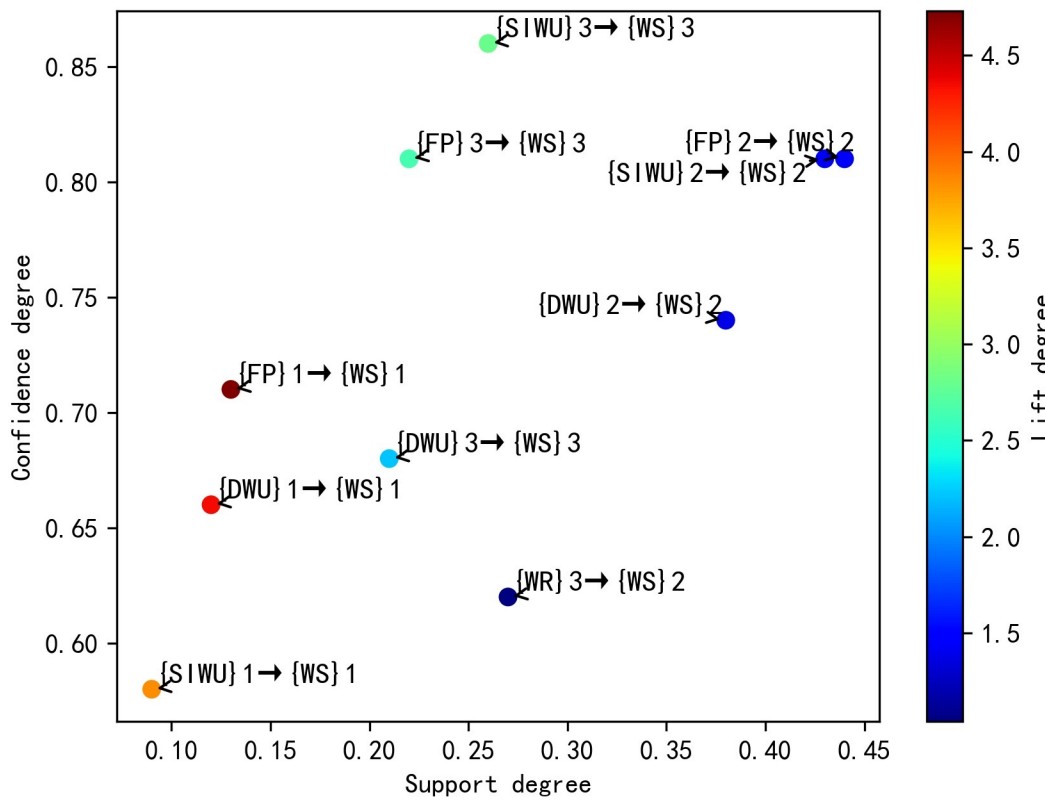

**Fig 4. The distribution plot of one-item valid SARs of D = 3.**

affect WS by affecting IWU. In the third category, FP is most likely to affect WS by affecting SIWU. Similarly, the WR and R affects WS through affecting SIWU and DUW, and WR and R are more likely to affect WS by affecting SIWU. Meanwhile, according to the rule 4, 7, 8 and 9, when R belongs to the first category, WS belongs to the third category. But when R belongs to the fourth category, WS still belongs to the third category, not the second category. This result shows that although R can affect WS by affecting SIWU and DUW, it has little influence on water supply fluctuation. Finally, the distribution plots of valid SARs are shown in Figs 4–7 respectively, and the color bars represent the association degree of SARs.

Through the above comparative analysis, it can be seen that the analysis results of different D have their own advantages. The average lift degree of valid SARs in D = 3 is higher than that of valid SARs in D = 4. Meanwhile, when D is equal to 3, the number of valid SARs is larger, and the sensitivity degree of features is higher. In the two-item SAR, the maximum lift degree of SARs in D = 4 was significantly lower than that of SARs in D = 3. Although there are spurious SARs in the results of D = 3, they can easily be filtered out by lift degree threshold, so the optimal value of D is 3.

## Discussion

In this study, the most critical parameters are support degree threshold and confidence degree threshold. We will present our experience in this section. Although the lift degree threshold is important, its definition already determines the threshold so that it does not need to be set manually.

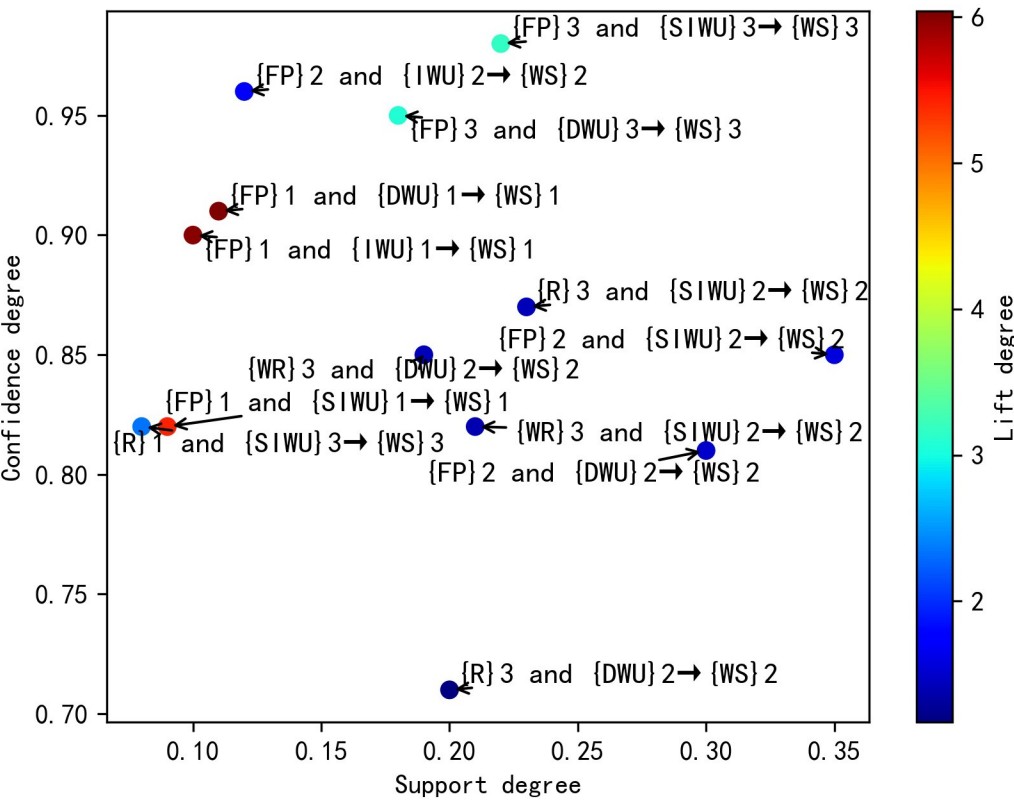

**Fig 5. The distribution plot of two-item valid SARs of D = 3.**

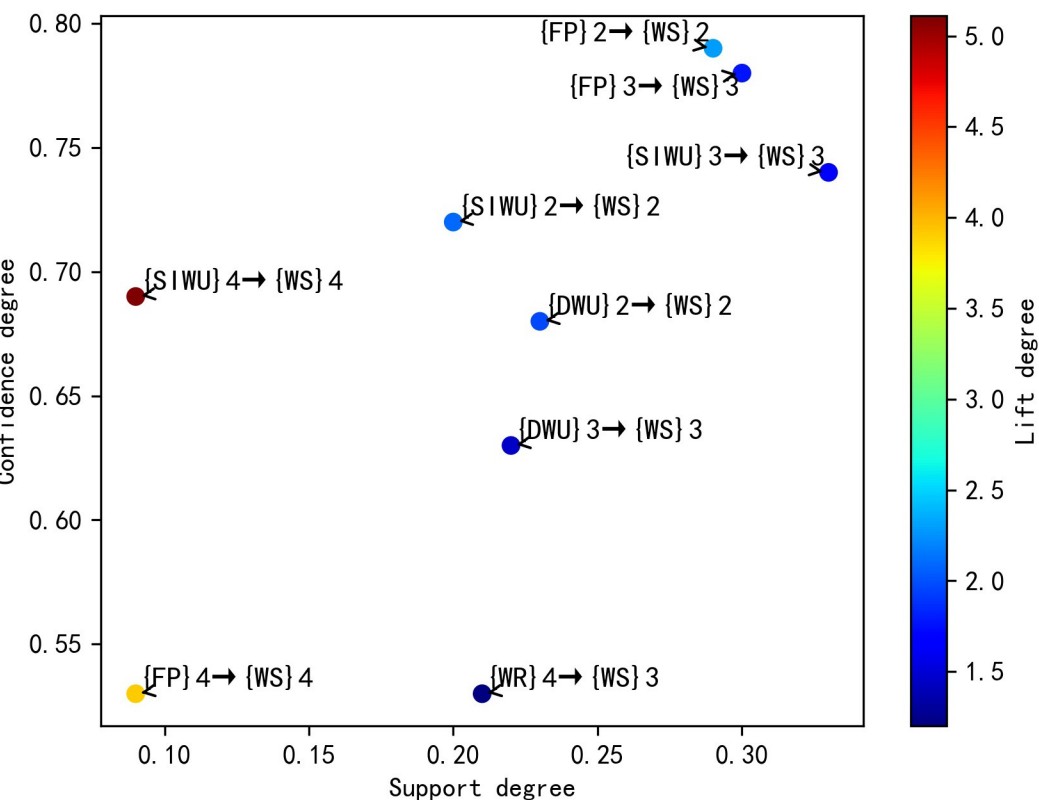

**Fig 6. The distribution plot of one-item valid SARs of D = 4.**

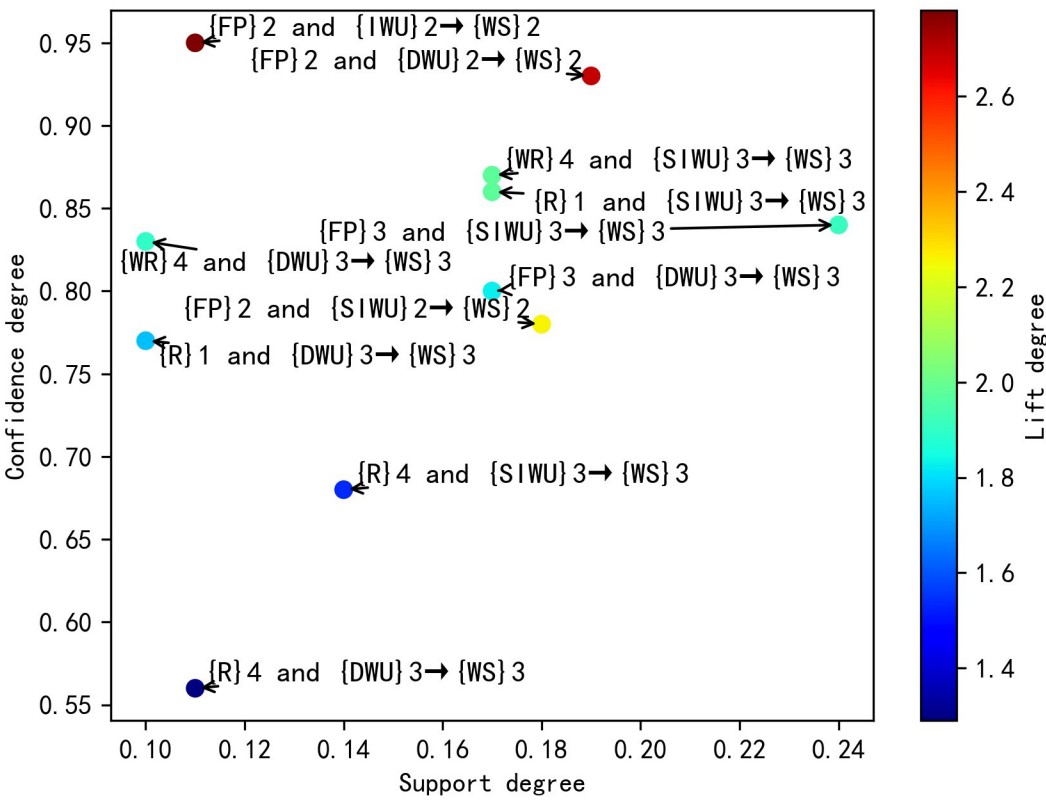

**Fig 7. The distribution plot of two-item valid SARs of D = 4.**

As mentioned in section 2.2 and 2.3, even though the *P(AB)* is small, it does not mean that the association degree between *A* and *B* is weak, this is because the number of total sample is very large. Therefore, the support degree threshold should be set as small as possible to prevent some potential SARs from being filtered out. Meanwhile, the support degree threshold and confidence degree threshold will affect the calculation load and efficiency of the algorithm, so the confidence degree threshold should not be set too small. In order to compare the data mining results of different thresholds, the confidence degree threshold is set to 0.1, while the support degree threshold remains unchanged. The data mining results are presented in Tables 9 and 10, and the SARs that appear in the results of section 4.2 is no longer displayed.

According to the six SARs in Table 9, it can be seen that only one SAR is valid and the lift degree of this SAR is 2.77. As can be seen from rules 1–3 in Table 5, the lift degree of the three SARs is higher than 2.77, indicating that the association degree between IWU and WS in the

**Table 9. The SAR difference of different confidence degree thresholds in D = 3.**

| Number | SAR | Support degree | Confidence degree | Lift degree |
|---|---|---|---|---|
| 1 | {IWU}2→{WS}2 | 0.15 | 0.49 | 0.91 |
| 2 | {IWU}1→{WS}1 | 0.12 | 0.42 | 2.77 |
| 3 | {IWU}3→{WS}3 | 0.12 | 0.29 | 0.96 |
| 4 | {R}1→{WS}3 | 0.10 | 0.18 | 0.50 |
| 5 | {R}1 and {WR}3→{WS}3 | 0.10 | 0.22 | 0.62 |
| 6 | {R}3 and {IWU}2→{WS}2 | 0.09 | 0.46 | 0.76 |

**Table 10. The SAR difference of different confidence degree thresholds in D = 4.**

| Number | SAR | Support degree | Confidence degree | Lift degree |
|---|---|---|---|---|
| 1 | {DWU}4→{WS}4 | 0.09 | 0.44 | 3.22 |
| 2 | {IWU}2→{WS}2 | 0.16 | 0.38 | 1.09 |
| 3 | {R}4→{WS}3 | 0.20 | 0.46 | 1.06 |
| 4 | {R}1→{WS}3 | 0.21 | 0.43 | 0.97 |

first category is not very strong. Therefore, the WS is more likely to be affected by FP, DWU and SIWU. Similarly, according to the results in Table 10, the lift degree of the first SAR is less than that of rules 1 and 2 in Table 7, and the lift degree of the second SAR is less than that of rules 3–5 in Table 7. Although the third SAR does not appear in Table 7, the lift degree of this SAR approaches 1, indicating that R and WS are almost independent of each other. Therefore, the data mining results with smaller confidence degree threshold have almost no influence on the results of this study, but it affects the calculation load and efficiency of the algorithm to some extent.

As can be seen from the second SAR in Table 7, the confidence degree of this SAR is 0.53, but the lift degree of this SAR is 3.91. In order to preserve the SAR that has a high lift degree, so the confidence degree threshold cannot be increased. The support degree threshold set in this study is already relatively small, and even if the support degree threshold is set to a smaller value, it just generates more frequent items. However, the association rules still need to be filtered by confidence degree threshold, so the data mining results of smaller support degree threshold are not presented in this study.

The value of D is inversely proportional to the width of the value interval. The smaller the width of the value interval, the easier it is to accurately determine the fluctuation interval of the WS. If we want to reduce the width of the value interval, we need to increase the D. The larger the value of D is, the less the number of valid SAR. At the same time, the maximum lift degree of the two-item SARs and feature sensitivity will decrease. Therefore, this is a classic problem of zero-sum game, and we have been devoted to finding the optimal scheme of D in zero-sum game.

Other analysis methods, such as these studies [42, 43], can only recognize sensitive features, but do not know how the features are associated with each other, and do not know whether their association relationship is direct or indirect. At the same time, the features recognized by these methods have not been validated. With the change of value interval of features, it is not known whether these sensitive features are still valid. Therefore, the analysis results of these methods are not reliable, and the method in this study is superior and more reliable.

## Conclusions

The purpose of this study is to carry out association analysis between features and WS through data mining to understand the cause of water supply fluctuation and the fluctuation interval of water supply, so as to realize attribution analysis of water supply fluctuation, and provide strong support for water supply dispatching. In this study, to improve the reliability of the analysis method, a data mining method coupling kmeans clustering discretization and apriori algorithm was proposed to carry out more reliable association analysis. The data discretization can avoid the influence of multicollinearity and monotone relationship during the analysis process. The scatter density plot matrix is used to intuitively discover the correlation relationship of data and explore the density distribution of the single feature and the two-dimensional density distribution between two features. The kmeans clustering algorithm is applied to carry

out the discretization of continuous data, and the apriori algorithm is used to carry out association analysis. The method in this study can not only obtain valid SARs which has been validated and the association degree of SAR, but also know the value interval of features. The results also show that the association relationship and association degree of the SAR is not completely unchanged, but closely associated with the value interval of features. This method in the study is a novel method for association analysis, which is more valid and reliable than current analysis methods. In addition, the study also provides guidance for avoiding the influence of multicollinearity and monotone relationship on the analysis results in the process of data analysis.

At present, the zero-sum problem of D has not been solved. We have been devoted to solving this problem. In particular, the number of SAR and the sensitivity of features should not be reduced while increasing the D. In the future, this research could be extended widely. The next step of the research work is to continue to design and develop discretization methods and analysis algorithms to compare their results. We can collect more data for association analysis to improve performance of algorithm, and use different languages instead of Python to discover the advantages of different languages in data mining. We will also use the methods of this study in other cities and compare the analysis results with those in Shenzhen.

## Author Contributions

**Conceptualization:** Jiaxuan Chang.

**Data curation:** Xin Liu.

**Formal analysis:** Xin Liu.

**Funding acquisition:** Xuefeng Sang.

**Investigation:** Xin Liu.

**Methodology:** Xin Liu.

**Supervision:** Xin Liu.

**Validation:** Xin Liu.

**Writing – original draft:** Xin Liu.

**Writing – review & editing:** Xin Liu, Xuefeng Sang, Jiaxuan Chang, Yang Zheng, Yuping Han.

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
