## [Decision Letter · Decision Letter 0]

9 Jun 2021

PONE-D-21-16186

The water supply association analysis method in Shenzhen based on kmeans clustering discretization and apriori algorithm

PLOS ONE

Dear Dr. Sang,

Thank you for submitting your manuscript to PLOS ONE. After careful consideration, we feel that it has merit but does not fully meet PLOS ONE’s publication criteria as it currently stands. Therefore, we invite you to submit a revised version of the manuscript that addresses the points raised during the review process.

We look forward to receiving your revised manuscript.

Kind regards,

Zaher Mundher Yaseen

Academic Editor

PLOS ONE

Journal Requirements:

a. You may seek permission from the original copyright holder of Figure(s) [#] to publish the content specifically under the CC BY 4.0 license.  

Reviewers' comments:

Reviewer's Responses to Questions

**Comments to the Author**

1. Is the manuscript technically sound, and do the data support the conclusions?

Reviewer #1: Yes

Reviewer #2: Yes

2. Has the statistical analysis been performed appropriately and rigorously? 

Reviewer #1: Yes

Reviewer #2: Yes

3. Have the authors made all data underlying the findings in their manuscript fully available?

Reviewer #1: Yes

Reviewer #2: Yes

4. Is the manuscript presented in an intelligible fashion and written in standard English?

Reviewer #1: Yes

Reviewer #2: Yes

5. Review Comments to the Author

Reviewer #1: 1. The manuscript presents the water supply association analysis method in Shenzhen based on kmeans clustering discretization and apriori algorithm, which is interesting. It is relevant and within the scope of the journal.

2. However, the manuscript, in its present form, contains several weaknesses. Appropriate revisions to the following points should be undertaken in order to justify recommendation for publication.

3. For readers to quickly catch your contribution, it would be better to highlight major difficulties and challenges, and your original achievements to overcome them, in a clearer way in abstract and introduction.

4. p.1 - a data mining coupling method is adopted for water supply attribution analysis. What are other feasible alternatives? What are the advantages of adopting this approach over others in this case? How will this affect the results? The authors should provide more details on this.

5. p.1 - kmeans clustering discretization is adopted to solve the optimal discretization degree to avoid multicollinearity problem. What are the other feasible alternatives? What are the advantages of adopting this soft computing technique over others in this case? How will this affect the results? More details should be furnished.

6. p.1 - apriori algorithm is adopted to solve the optimal discretization degree to obtain association rule and association degree. What are the other feasible alternatives? What are the advantages of adopting this algorithm over others in this case? How will this affect the results? More details should be furnished.

7. p.2 - Shenzhen is adopted as the case study. What are other feasible alternatives? What are the advantages of adopting this case study over others in this case? How will this affect the results? The authors should provide more details on this.

8. p.3 - historical records of 2004 to 2019 are taken. Why are more recent data not included in the study? Is there any difficulty in obtaining more recent data? Are there any changes to situation in recent years? What are its effects on the result?

9. p.3 - the discretization results of three methods as shown in Fig. 2 are adopted in this study What are the other feasible alternatives? What are the advantages of adopting these methods over others in this case? How will this affect the results? More details should be furnished.

10. p.5 - Eq. 10 is adopted as the objective function. What are other feasible alternatives? What are the advantages of adopting this function over others in this case? How will this affect the results? The authors should provide more details on this.

11. p.5 - Eqs. 11 and 12 are adopted as the minimum support degree thresholds. What are the other feasible alternatives? What are the advantages of adopting these thresholds over others in this case? How will this affect the results? More details should be furnished.

12. p.6 - six features are adopted in the study. What are the other feasible alternatives? What are the advantages of adopting these features over others in this case? How will this affect the results? More details should be furnished.

13. p.6 - spearman's rank correlation coefficient is adopted to select the features. What are the other feasible alternatives? What are the advantages of adopting this approach over others in this case? How will this affect the results? More details should be furnished.

14. p.6 - a scatter density diagram matrix is adopted to guide subsequent analysis. What are the other feasible alternatives? What are the advantages of adopting this approach over others in this case? How will this affect the results? More details should be furnished.

15. p.8 - “…Compared with D = 4, D = 3 is considered to be better in this study, because the.…” More justification should be furnished on this issue.

16. p.8 - SAR with one-item set and two-item set are adopted in the experiments. What are the other feasible alternatives? What are the advantages of adopting this approach over others in this case? How will this affect the results? More details should be furnished.

17. Some key parameters are not mentioned. The rationale on the choice of the particular set of parameters should be explained with more details. Have the authors experimented with other sets of values? What are the sensitivities of these parameters on the results?

18. Some assumptions are stated in various sections. Justifications should be provided on these assumptions. Evaluation on how they will affect the results should be made.

19. The discussion section in the present form is relatively weak and should be strengthened with more details and justifications.

20. Moreover, the manuscript could be substantially improved by relying and citing more on recent literatures about contemporary real-life case studies of modelling and/or optimization techniques in water distribution systems such as the followings. Discussions about result comparison and/or incorporation of those concepts in your works are encouraged:

Zheng FF, et al. “Improved Understanding on the Searching Behavior of NSGA-II Operators Using Run-Time Measure Metrics with Application to Water Distribution System Design Problems,” Water Resources Management 31 (4): 1121-1138 2017.

Shende S, et al. “Design of water distribution systems using an intelligent simple benchmarking algorithm with respect to cost optimization and computational efficiency,” Water Supply 19 (7): 1892-1898 2019.

Sedki A, et al. “Hybrid particle swarm optimization and differential evolution for optimal design of water distribution systems,” Advanced Engineering Informatics 26 (3): 582-591 2012.

Oyebode O, et al. “Evolutionary modelling of municipal water demand with multiple feature selection techniques,” Journal of Water Supply Research and Technology-AQUA, 68 (4): 264-281 2019

21. Some inconsistencies and minor errors that needed attention are:

Replace “…So Compared with D = 4, D = 3 is considered to be…” with “…So compared with D = 4, D = 3 is considered to be…” in p.8

22. In the conclusion section, the limitations of this study and suggested improvements of this work should be highlighted.

Reviewer #2: The manuscript entitled “The water supply association analysis method in Shenzhen based on kmeans clustering discretization and apriori algorithm” looks an applied research based mainly on the kmeans clustering and apriori algorithm that had been applied on Shenzhen, a city located to south of China. It was well structured and written in good way. However I recommend a major revision to address the following points:

Abstract

The research abstract seems almost vague as if it is a fragmented part of the research. The abstract should be reformulated to be easy to understand and expressive of the overall research content.

Introduction

I think the introduction was written in a format that could be described as good, but it still did not live up to the preferences. It is better to divide the introduction to include the scientific background of the main topic and the methods of work used in the research. After that, the research problem should be stated clearly and the reserch importance addressed. Then the previous studies in the field of research are listed. Accordingly, the introduction should be written in a more effective way to cover all aspects of the topic. The subject of the introduction focused on aspects and overlooked other important aspects. Some mathematical methods have been presented that can contribute to establishing a certain guess, while the water supply itself is not given any importance in terms of clarifying the factors controlling it in theory. The research problem should be clearly clarified in a separate sentence, and then presenting the originality of the idea and the importance of the research, what is the intended final benefit, and whether the research aims at any knowledge addition at the academic or applied research level. What makes the research subject to criticism is not to address more previous studies and present their most important conclusions. It is also preferable to redesign the target and write it in a clear text, away from ambiguity. I would prefer to re-state the aim of study “This study can establish valid SAR among features and recognize association degree of SAR. Finally, the value range of features in SAR can be obtained through the association rules”.

Materials and methods

- The situation of study area was included within the materials and methods section! It is not suitable to be placed in this section; it is better to be as an independent section.

- At last paragraph of the Materials and Methods, the author has stated “The data for this study came from the monthly data of Shenzhen Bureau of Water, the Shenzhen Bureau of Statistics, the Shenzhen Water Group and Digital Water System from 2004 to 2019. The methods in this study were developed in Python3”. This topic is not related directly to the study area but rather to the methods of work and data collection, so I would prefer to transfer it to suitable place.

What colors (green, orange, and blue) mean in Fig. 2.?

I do not understand why the author resorts to drawing equations for some common topics, and if he referred to computer software programs, it would be better, for example “spearman's rank correlation coefficient”.

The scatter diagrams need to be explained how clarifies the relationships between different features.

The expression 3D = 3 in section 2.2. Mining results and discussion: is not accepted to be written by this style, please.

The highest confidence degree in Table 3 is low (0.7)? Can such a result be taken into consideration?

6. PLOS authors have the option to publish the peer review history of their article (what does this mean?). If published, this will include your full peer review and any attached files.

Reviewer #1: No

Reviewer #2: **Yes: **Salih Muhammad Awadh

---

## [Author Response · Author response to Decision Letter 0]

5 Jul 2021

Journal Requirements:

1. If applicable, we recommend that you deposit your laboratory protocols in protocols.io to enhance the reproducibility of your results.

Answer: the protocol has been created. Since our experiments were done separately, and we might apply for a patent based on this method, so we only uploaded data set and code demonstration. However, the code can run smoothly and generate the results, and the same is true for other categories of association analysis. https://www.protocols.io/workspaces/water-supply-association-analysis

2. Ensure that your manuscript meets PLOS ONE's style requirements, including those for file naming.

Answer: we agree to revise. The style in the paper had been revised.

3. The grant information you provided in the ‘Funding Information’ and ‘Financial Disclosure’ sections do not match. 

Answer: we have checked it carefully. Based on Submission Guidelines, the funding sources in Acknowledgments have been removed. The National Key Research and Development Program of China and National Natural Science Foundation of China have grant numbers, but the Innovation Foundation of North China University of Water Resources and Electric Power for PhD Graduates do not have a grant number, because this Innovation Foundation is actually an honor institution established by the university. The fund is provided to outstanding doctors for scientific research and innovation. Although it is named as Innovation Foundation, it is not an official organization. It is just an honor organization, so there is no grant number. But the money does go to research, so the university requires that the funding information should be included in the paper.

4. We note that Figure 1 in your submission contain [map/satellite] images which may be copyrighted.

Answer: we agree to revise. The main content of this study is data mining, which is not closely related to the location of the study area, so Figure 1 has been removed. 

For the comments of reviewer #1

The first comment and the second comment are statements by the reviewer, so they do not need to be revised.

1. 3. For readers to quickly catch your contribution, it would be better to highlight major difficulties and challenges, and your original achievements to overcome them, in a clearer way in abstract and introduction.

Answer: we agree to revise. We rewrote the abstract and introduction. Major difficulties, challenges and the original achievements to overcome them have been stated in the abstract and introduction. We have also revised some inappropriate use and language in the abstract and introduction.

2. 4. A data mining coupling method is adopted for water supply attribution analysis. What are other feasible alternatives? What are the advantages of adopting this approach over others in this case? How will this affect the results? 

Answer: Unfortunately, through extensive experiments and researches, we have not yet developed a feasible alternative. However, we believe that as long as an algorithm can be analyzed based on discrete data and the analysis results can be validated by certain means, so this method is feasible alternative. We have taken this problem as our current key research direction, hoping to find another effective data mining method of association rule, and compare the two methods to find advantages and disadvantages of different methods. At the same time, we will try different computing languages to compare the advantages of languages in data mining. 

The mainstream analysis methods can only recognize sensitive features, but do not know how the features are associated with each other, and do not know whether their association relationship is direct or indirect. At the same time, the features recognized by these methods have not been validated. With the change of value interval of features, it is not known whether these sensitive features are still valid. The data mining method based on discrete data proposed in this study solves the above problems, obtains association rules and filters out invalid rules through validation. In addition, we can obtain the value range of features in rules, the analysis results are reliable. 

Therefore, the results of these mainstream analysis methods are not reliable, and the method in this study is superior and more reliable. 

These contents have been presented in the introduction, results and conclusions.

3. 5. The kmeans clustering discretization is adopted to solve the optimal discretization degree to avoid multicollinearity problem. What are the other feasible alternatives? What are the advantages of adopting this soft computing technique over others in this case? How will this affect the results?

Answer: The commonly used discretization methods include equal width discretization, equal frequency discretization and clustering discretization. Their discretization results and defect have been described in section 2.1 (Discretization) of the paper. The quality of discretization interval directly affects the quality of the analysis results. The bad discretization interval may cause the algorithm to miss some valuable rules. Through experiments, it is found that the discretization results of clustering method are optimal. Kmeans is recognized as a method with good effect and low computation load among the current cluster analysis methods, and this method can generate interval breakpoints and has self-adaptive ability. 

Other methods cannot map the analysis results to a certain interval. The research methods iteratively carry out the algorithm for different intervals, and the association rules of different association degrees can be mapped to different sensitive intervals, making the analysis results more clear. Kmeans algorithm is self-adaptive, and we can also manually set the parameters of the algorithm, such as random state and maximum iteration times of the algorithm, which is also one of the reasons for the good analysis effect.

These contents have been revised in section 2.1 (Discretization) of the paper.

4. 6. The apriori algorithm is adopted to solve the optimal discretization degree to obtain association rule and association degree. What are the other feasible alternatives? What are the advantages of adopting this algorithm over others in this case? How will this affect the results?

Answer: the other feasible alternative is fpgrowth algorithm. For fpgrowth algorithm, events are mapped to a path in the FP tree to construct the tree structure. Although the two algorithms are different, but the data mining results is the same, so we're not going to repeat it.

As for mainstream association methods, such as clustering, neural network, similarity, time series, can not effectively obtain the analysis results in this study. Compared with fpgrowth algorithm, the apriori algorithm adopts the iterative method of searching layer by layer, the algorithm is simple and has better performance, and the algorithm is easy to be developed.

The advantages of adopting this algorithm have been stated in section 1 (Introduction) and section 2 (Materials and methods).

5. 7. Shenzhen is adopted as the case study. What are other feasible alternatives? What are the advantages of adopting this case study over others in this case? How will this affect the results? The authors should provide more details on this.

Answer: the research method is data-driven, so any city that can provide detailed data can be used as case study. 

Our team is participating in the construction of the first stage of Shenzhen Wisdom Water Project, which includes attribution analysis of water supply fluctuations. Shenzhen is a prototype of socialist modern city construction and Shenzhen is a pilot city of wisdom water and intelligent dispatching in China. So there's a lot of monitoring devices and metering devices in Shenzhen, and there's also big data systems and digital water system that allow us to get detailed data. Wisdom water is being gradually promoted in China, and we will apply the research methods to cities that can provide detailed data. Meanwhile, we really hope to use the detailed data provided by other cities for analysis and compare with the analysis results of Shenzhen to find the advantages. We also hope that this research method can provide support for the water supply of Shenzhen and other cities in China. 

The data mining methods can find the hidden information that the data cannot tell us, and obtain the previously unknown and valuable knowledge. But it is impossible to carry out data mining without enough data

This part of the content is not relevant to the research, so it was not revised in the paper.

6. 8. Historical records of 2004 to 2019 are taken. Why are more recent data not included in the study? Is there any difficulty in obtaining more recent data? Are there any changes to situation in recent years? What are its effects on the result?

Answer: The coming period is a critical period for Shenzhen to build socialist modernization, a crucial period for implementing the construction of the Guangdong-Hong Kong-Macao Greater Bay Area. Shenzhen is also a special economic zone in China, and the water supply of Hong Kong mainly relies on the Shenzhen Reservoir. Therefore, the recent measured data was considered confidential, and our team was unable to obtain the recent measured data.

Shenzhen is the first city in China to be fully urbanized, and it has complete water, land, air, and railway ports. Therefore, the urban functions are complete, so there are no big changes in Shenzhen in recent years.

Data mining method is a data-driven method that find valuable information from data, so changes in the external environment will not affect the data mining method.

This part of the content is not relevant to the research, so it was not revised in the paper.

7. 9. The discretization results of three methods as shown in Fig. 2 are adopted in this study What are the other feasible alternatives? What are the advantages of adopting these methods over others in this case? How will this affect the results? More details should be furnished.

Answer: the other feasible alternatives are supervised methods. The data discretization methods include supervised methods and unsupervised methods, and the classification criterion is whether the data contains category information. The supervised discretization takes into account category information while unsupervised discretization does not. If supervised discretization method is to be used, manual annotation of data is needed to add category information, which is very complicated, and the manual annotation results may have many kinds. Therefore, unsupervised discretization methods are used more widely, and the unsupervised discretization methods are selected in this study. The unsupervised discretization methods include equal width discretization, equal frequency discretization and clustering discretization. 

The quality of discretization interval directly affects the quality of the analysis results. The bad discretization methods may cause damage in the data mining process.

These contents have been revised in the section 2.1 (Discretization).

8. 10. Eq. 10 is adopted as the objective function. What are other feasible alternatives? What are the advantages of adopting this function over others in this case? How will this affect the results? The authors should provide more details on this.

Answer: Since the objective function is set according to the idea of the researcher, we believe that as long as the function is reasonable, it can be used as an alternative.

Since association rules obtained by data mining methods need to be validated, the purpose of this study is not only to obtain more association rules, but also to obtain more effective association rules. We may lose quality if we focus only on quantity of SAR, and We may lose quantity if we focus only on quality of SAR. Therefore, the objective function set in this study is that the average lift degree of the valid association rules is maximum. This objective function makes the analysis results of iterative calculation more reliable, so we can get a more reliable and universal method.

These contents have been revised in the section 2.3 (Coupling method).

9. 11. Eqs. 11 and 12 are adopted as the minimum support degree thresholds. What are the other feasible alternatives? What are the advantages of adopting these thresholds over others in this case? How will this affect the results? More details should be furnished.

Answer: the number of feasible alternatives is enormous, because this threshold can be set manually. 

Threshold is also need to meet several criteria, such as computation load and efficiency. If the support degree threshold is set too high, although it can reduce the time to calculate frequent item sets in data mining, it is easy to cause some association items hidden in the data to be filtered out. Because the confidence degree need to be calculated after support degree, so the support degree threshold should be set as small as possible. If the confidence degree threshold is set too low, a large number of invalid rules may be generated, leading to a high calculation load and greatly increasing the time of data mining. Therefore, Therefore, this study gives consideration to both calculation load and efficiency, and combined with previous data mining experience to finally set these thresholds.

In addition, we also set a small confidence degree threshold in the results and discussion to compare the differences in the analysis results. Because no matter how the support degree threshold is set, rules need to be filtered by confidence degree threshold, so this study only discusses the difference of the confidence degree threshold. 

These contents have been revised in section 2.3 (Coupling method), section 4.2 (Experiment results) and section 5 (Discussion).

10. 12. Six features are adopted in the study. What are the other feasible alternatives? What are the advantages of adopting these features over others in this case? How will this affect the results? More details should be furnished.

Answer: the water use of Shenzhen consists of domestic water use, industrial water use, service industry water use, ecological water use, agricultural water use and construction industry water use. The other feasible alternatives include ecological water use, agricultural water use and construction industry water use. 

The domestic water use, industrial water use, service industry water use were selected by spearman's rank correlation coefficient. Shenzhen is a fully urbanized city with a floating population of more than 8 million. Shenzhen has abundant rainfall, with an average annual rainfall of 1830 mm. At the same time, a large amount of water use has produced a large amount of wastewater, so the amount of wastewater reuse in Shenzhen is increasing. The six features have great influence on water supply, and at the same time, these features are more targeted. Other features have little relationship with water supply fluctuations, so they cannot generate valid strong association rule between them and water supply. At the same time, we can reduce the calculation load of data mining by not choosing them.

This has been added in section 3.2 (Data description).

11. 13. Spearman's rank correlation coefficient is adopted to select the features. What are the other feasible alternatives? What are the advantages of adopting this approach over others in this case? How will this affect the results? More details should be furnished.

Answer: the other feasible alternative is Pearson correlation coefficient. The Pearson correlation coefficient is an evaluation method of the linear relationship between two variables. When the variables do not follow a normal distribution or have more complex linear correlation degree, the Pearson coefficient is no longer valid. Spearman's rank correlation coefficient evaluates a monotone relationship rather than a linear relationship between two variables. Spearman coefficient does not require prior knowledge, and its application scope is wider than Pearson coefficient.

The analysis results of different correlation coefficient may be different. This study focuses on the attribution analysis of water supply and searching for valid strong association rules, so it is not suitable to use linear relationship as the standard.

This has been added in Section 3.2 (Data description).

12. 14. A scatter density diagram matrix is adopted to guide subsequent analysis. What are the other feasible alternatives? What are the advantages of adopting this approach over others in this case? How will this affect the results? More details should be furnished

Answer: the other alternatives include Pearson correlation coefficient and Spearman's rank correlation coefficient, which can also be used to describe the relationship between variables. But the relationships they can describe are very limited. 

The scatter density plot matrix can describe a variety of complex relationships, such as the density distribution of the single feature, the density relationship and correlation relationship between features, and the function of this method is strong.

The exploratory data analysis (EDA) can let us intuitively understand the characteristics of data and know the potential relationship between them, which is a valid method of data analysis. Meanwhile, EDA helps discover what the data is trying to tell us and can be used to look for patterns and relationships to guide our subsequent analysis.

This has been added in section 4.1 (Exploratory data analysis).

13. 15. “…Compared with D = 4, D = 3 is considered to be better in this study, because the.…” More justification should be furnished on this issue.

Answer: we agree to revise. The content has been added to section 4.2 (Experiment results).

14. 16. SAR with one-item set and two-item set are adopted in the experiments. What are the other feasible alternatives? What are the advantages of adopting this approach over others in this case? How will this affect the results? More details should be furnished.

Answer: the other feasible alternatives are three-item SAR and four-item SAR. 

This is because the feature strongly associated with WS has dominated the SAR, which is equivalent to give this feature a large weight. If a new feature which has a strong association degree with WS is added to the two-item SAR, the three-item SAR will undoubtedly be has a strong association degree with WS. Even if a feature that is completely uncorrelated with WS is added to the two-item SAR, the three-item SAR still has a strong association degree with WS due to the existence of the feature which has a large weight. Take the first rule in Table 6 of the paper as an example, the lift degree is very high, so the association degree of the rule is very high. For example, if SIWU is added, there is no doubt that the three-item rule is a valid strong association rule with water supply. If R is added, although the association degree between R and WS is not strong, the FP and DWU have very strong association degree with water supply. So the addition of R is equivalent to adding some noise to the previous SAR, which will only have a weak impact on the analysis results. However, this result is not consistent with our original intention. Therefore, this study only focuses on the one-item SAR and two-item SAR, which can also reduce the calculation load of data mining. 

This part has been added in Section 4.2 (Experiment results).

15. 17. Some key parameters are not mentioned. The rationale on the choice of the particular set of parameters should be explained with more details. Have the authors experimented with other sets of values? What are the sensitivities of these parameters on the results?

Answer: we agree to revise. A more detailed statement has been added to the paper. In this study, we had experimented with other sets of values, and the results showed that the rules of three-item and four-item set had certain association degree with WS. But this study only focuses on the one-item SAR and two-item SAR, which can also reduce the calculation load of data mining. The reasons had been stated in the question 16, these results are not consistent with our original intention, so this study did not focus on them.

16. 18. Some assumptions are stated in various sections. Justifications should be provided on these assumptions. Evaluation on how they will affect the results should be made.

Answer: all the results in the paper are the real results obtained from data mining without assumptions.

17. 19. The discussion section in the present form is relatively weak and should be strengthened with more details and justifications.

Answer: we agree to revise. The results and discussion have been strengthened. At the same time, the section 5 (Discussion) is added, which discusses the results of different confidence degree threshold. The support degree threshold set in this study is already relatively small, and even if the support degree threshold is set to a smaller value, it just generates more frequent items. However, the association rules still need to be filtered by confidence degree threshold, so the data mining results of smaller support degree threshold are not presented in this study.

18. 20. Moreover, the manuscript could be substantially improved by relying and citing more on recent literatures about contemporary real-life case studies of modelling and/or optimization techniques in water distribution systems such as the followings. Discussions about result comparison and/or incorporation of those concepts in your works are encouraged.

Answer: we agree to revise. We have studied these papers carefully and decided to cite them.

19. 21. Some inconsistencies and minor errors that needed attention are: Replace “…So Compared with D = 4, D = 3 is considered to be…” with “…So compared with D = 4, D = 3 is considered to be…” in p.8

Answer: we agree to revise. This capital letter has been revised.

20. 22. In the conclusion section, the limitations of this study and suggested improvements of this work should be highlighted.

Answer: we agree to revise. This part has been added in conclusions.

For the comments of reviewer #2

21. 1. The research abstract seems almost vague as if it is a fragmented part of the research. The abstract should be reformulated to be easy to understand and expressive of the overall research content.

Answer: we agree to revise. The abstract is rewritten and revised further.

22. 2. I think the introduction was written in a format that could be described as good, but it still did not live up to the preferences. It is better to divide the introduction to include the scientific background of the main topic and the methods of work used in the research. 

Answer: we agree to revise. The structure of the introduction was tweaked and rewritten.We clearly stated the importance and expected benefits of this study, analyzed the limitations of previous studies, and clearly stated the objectives of this study.

23. 3. Materials and methods. The situation of study area was included within the materials and methods section! It is not suitable to be placed in this section; it is better to be as an independent section. At last paragraph of the Materials and Methods, the author has stated “…”. This topic is not related directly to the study area but rather to the methods of work and data collection, so I would prefer to transfer it to suitable place.

Answer: we agree to revise. The study area and data description will be placed in section 3 (Study area and data description), which is an independent section. The development tools of the algorithm are placed in section 2.3 (Coupling method), the statement of the coupling method.

24. 4.What colors (green, orange, and blue) mean in Fig. 2.?

Answer: these three colors represent the value distribution of in the interval after discretization. It has been revised in the paper to add detailed description in section 2.1 (Discretization).

25. 5. I do not understand why the author resorts to drawing equations for some common topics, and if he referred to computer software programs, it would be better, for example “Spearman's rank correlation coefficient”.

Answer: The algorithm and parameters in this study are obtained through previous data mining experience and a large number of experiments. The explanation and statement of the methods will to some extent weaken the importance of the topic. The algorithm, flow of methods and parameters are expressed by the equation more clearly and intuitively. Spearman's rank correlation coefficient is directly used in this study to calculate the monotone relationship of features, but this method is not developed in this study. Meanwhile, this method is so common that it doesn't need to be stated or explained.

26. 6. The scatter diagrams need to be explained how clarifies the relationships between different features.

Answer: we agree to revise. The more detailed statement clarifying the relationships between different features has been added to section 4.1 (Exploratory data analysis).

27. 7. The expression 3D = 3 in section 2.2. Mining results and discussion: is not accepted to be written by this style, please.

Answer: we agree to revise. But I don't understand the meaning of the sentence "The expression 3D = 3 in section 2.2.". I don't find the expression 3D=3 in section 2.2. 

Mining results and discussion: the title has been revised. 

28. 8. The highest confidence degree in Table 3 is low (0.7)? Can such a result be taken into consideration?

Answer: the rules in the table are sorted by lift degree, and the highest confidence degree is not 0.7. As long as the confidence degree of the rule is greater than the confidence degree threshold, they should all be considered. Finally, the rule must be validated by the lift degree. If the lift degree of the rule is greater than 1, the rule is considered to be valid. The association degree of rule is relevant depends on the lift degree. Therefore, as long as the rules meet the threshold conditions, they should all be considered.

Self-revision

The variable description (Table 1) is added in the section 3.2 (Data description).

The discretization results of D = 3 and the discretization results of D = 4 are presented in two tables, respectively. The maximum and minimum values of the features are added to the value interval.

In the process of repeatedly reading and revising the paper, we found some mistakes and some unclear expressions, and these problems have been revised.

---

## [Decision Letter · Decision Letter 1]

22 Jul 2021

The water supply association analysis method in Shenzhen based on kmeans clustering discretization and apriori algorithm

PONE-D-21-16186R1

Dear Dr. Sang,

We’re pleased to inform you that your manuscript has been judged scientifically suitable for publication and will be formally accepted for publication once it meets all outstanding technical requirements.

Kind regards,

Zaher Mundher Yaseen

Academic Editor

PLOS ONE

Additional Editor Comments (optional):

Reviewers' comments:

Reviewer's Responses to Questions

**Comments to the Author**

1. If the authors have adequately addressed your comments raised in a previous round of review and you feel that this manuscript is now acceptable for publication, you may indicate that here to bypass the “Comments to the Author” section, enter your conflict of interest statement in the “Confidential to Editor” section, and submit your "Accept" recommendation.

Reviewer #1: All comments have been addressed

Reviewer #2: All comments have been addressed

2. Is the manuscript technically sound, and do the data support the conclusions?

Reviewer #1: Yes

Reviewer #2: Yes

3. Has the statistical analysis been performed appropriately and rigorously? 

Reviewer #1: Yes

Reviewer #2: Yes

4. Have the authors made all data underlying the findings in their manuscript fully available?

Reviewer #1: Yes

Reviewer #2: Yes

5. Is the manuscript presented in an intelligible fashion and written in standard English?

Reviewer #1: Yes

Reviewer #2: Yes

6. Review Comments to the Author

Reviewer #1: (No Response)

Reviewer #2: Through my review of this round of scientific evaluation of the manuscript entitled “The water supply association analysis method in Shenzhen based on kmeans clustering discretization and apriori algorithm” and after reviewing the author's response to the comments of the reviewers. I found that the author has responded very well and submitted a satisfactory modified version that meets the research basics. The modified manuscript becomes more enhanced than before, and on this basis, Eventually, the author has adequately addressed the reviewer comments raised in a previous round of review and I feel the manuscript is now acceptable for publication. I here authorize the editor to accept it as my decision is to accept the research.

7. PLOS authors have the option to publish the peer review history of their article (what does this mean?). If published, this will include your full peer review and any attached files.

Reviewer #1: No

Reviewer #2: **Yes: **Salih Muhammad Awadh

---

## [Editor Report · Acceptance letter]

27 Jul 2021

PONE-D-21-16186R1 

The water supply association analysis method in Shenzhen based on kmeans clustering discretization and apriori algorithm 

Dear Dr. Sang:

I'm pleased to inform you that your manuscript has been deemed suitable for publication in PLOS ONE. Congratulations! Your manuscript is now with our production department. 

Kind regards, 

on behalf of

Dr. Zaher Mundher Yaseen 

Academic Editor

PLOS ONE